# Triple-negative breast cancer influences a mixed M1/M2 macrophage phenotype associated with tumor aggressiveness

**Kristine Cate S. Pe**[1], **Rattana Saetung**[1], **Varalee Yodsurang**[1], **Chatchai Chaotham**[2], **Koramit Suppipat**[3,4], **Pithi Chanvorachote**[1], **Supannikar Tawinwung**[1,4]*

**1** Faculty of Pharmaceutical Sciences, Department of Pharmacology and Physiology, Chulalongkorn University, Bangkok, Thailand, **2** Faculty of Pharmaceutical Sciences, Department of Biochemistry and Microbiology, Chulalongkorn University, Bangkok, Thailand, **3** Faculty of Medicine, Department of Research Affair, Chulalongkorn University, Bangkok, Thailand, **4** Cellular Immunotherapy Research Unit, Chulalongkorn University, Bangkok, Thailand

* supannikar.T@pharm.chula.ac.th

**Data Availability Statement:** All relevant data are within the paper and its Supporting Information files.

## Abstract

Triple-negative breast cancer (TNBC) is characterized by excessive accumulation of tumor-infiltrating immune cells, including tumor-associated macrophages (TAMs). TAMs consist of a heterogeneous population with high plasticity and are associated with tumor aggressiveness and poor prognosis. Moreover, breast cancer cells can secrete factors that influence TAM polarization. Therefore, this study aimed to evaluate the crosstalk between cancer cells and macrophages in the context of TNBC. Cytokine-polarized M2 macrophage were used as control. Distinct from the classical M2 macrophage, TAMs generated from TNBC-conditioned media upregulated both M1- and M2-associated genes, and secreted both the anti-inflammatory cytokine interleukin IL-10 and the proinflammatory cytokine IL-6 and tumor necrosis factor- α. Theses TNBC-induced TAMs exert aggressive behavior of TNBC cells. Consistently, TCGA and MTABRIC analyses of human breast cancer revealed upregulation of M1- associated genes in TNBC comparing with non-TNBC. Among these M1-associated genes, *CXCL10* and *IL1B* were revealed to be independent prognostic factors for disease progression. In conclusion, TNBC cells induce macrophage polarization with a mixture of M1 and M2 phenotypes. These cancer-induced TAMs further enhance tumor cell growth and aggressiveness.

## Introduction

Triple-negative breast cancer (TNBC) is the most aggressive subtype of breast cancer, characterized by negative expression of the estrogen receptor, progesterone receptor, and HER2. Patients with TNBC have a relatively poor prognosis, owing to an aggressive tumor behavior and the lack of molecular targets for therapy [1,2]. In addition, TNBC features a unique microenvironment distinct from other breast cancer subtypes, including a high level of tumor-infiltrating lymphocytes and tumor-associated macrophages (TAMs) [3]. TAMs originate from peripheral blood monocytes and differentiate into macrophages following recruitment to

**Funding:** ST MRG6180200 the Thailand Research Fund http://academics.trf.or.th/ No The funders had no role in study design, data collection and analysis, decision to publish, or preparation of the manuscript.

**Competing interests:** The authors have declared that no competing interests exist.

tumor sites. Accumulated evidence suggests that TAMs are a heterogeneous and plastic population, in which polarized TAMs can be identified as M1- and M2-like macrophages [4]. M1-like macrophages are known to release proinflammatory cytokines and chemokines such as tumor necrosis factor (TNF)-α, interleukin [IL]-1β, and CXCL10 and exert antitumor activity, whereas M2-like macrophages produce anti-inflammatory cytokines such as IL-4, IL-10, and IL-13 and are associated with tissue remodeling, angiogenesis, and immune suppression, leading to tumor development [5,6].

Generally, TAMs exhibit characteristics similar to those of anti-inflammatory M2 macrophages, which enhance the aggressive features of several cancers [7]. Primary TNBC tissues express high levels of CD163+ M2-like macrophages, and the expression of CD163+ in the tumor stroma is associated with the absence of hormone receptors and increased aggressive features of breast cancer [8,9]. Previous studies have also reported that TNBC cells educate primary monocytes toward the M2-like macrophage phenotype by secreting high amounts of monocyte colony-stimulating factor (M-CSF) [8,10]. By contrast, TAMs can secrete soluble factors including growth factors, cytokines, and chemokines such as transforming growth factor [TGF]-β, vascular endothelial growth factor, M-CSF, IL-10, and CXCL10 to induce tumor progression and metastasis [11–14].

In this study, we aimed to evaluate the crosstalk between TNBC cells and macrophages. We found that soluble factors obtained from a TNBC cell line modulate monocytes toward a mixed population of M1- and M2-like macrophages. Unlike classical M2-polarized macrophage, these TAMs show an upregulation of M1-associated genes and secrete the proinflammatory cytokine IL-6. TNBC cells co-cultured with TAMs exhibited increased cell proliferation and migration. Thus, our results support the complex interactions between macrophages and TNBC in tumor development.

## Materials and methods

### Cell culture

The protocol and written informed consent for obtaining peripheral blood from healthy donors were approved by the Institutional Review Board of the Faculty of Medicine, Chulalongkorn University (IRB NO.437/62). The MDA-MB-231, MDA-MB-468, and MCF-7 human breast cancer cell lines and THP-1 monocytic cell lines, obtained from the American Type Culture Collection (ATCC®, VA, USA), were grown in Dulbecco's Modified Eagle Medium (DMEM) (Gibco, NY, USA) and Roswell Park Memorial Institute (RPMI)-1640 medium (Gibco), respectively, at 37˚C in a humidified incubator with 5% carbon dioxide. Both media were supplemented with 10% fetal bovine serum (FBS), 1% penicillin/streptomycin, and 1X GlutaMAX (Gibco).

### Macrophage differentiation

THP-1 cells were stimulated with 100 nM phorbol 12-myristate 13-acetate (PMA) (Sigma-Aldrich, Saint Louis, MO, USA) in RPMI medium for 24 h, followed by a 24-h rest period in fresh RPMI-1640 media to allow differentiation into macrophages. For the generation of TAMs, THP-1-derived macrophages were treated with 2 mL of conditioned media (CM) from breast cancer cells for 48 h. For M2-like macrophage polarization, IL-4 and IL-13 were added to fresh RPMI media for 48 h at a final concentration of 20 ng/mL and 20 ng/mL, respectively.

### Breast cancer cell CM preparation

The TNBC cell lines MDA-MB-231 and MDA-MB-468 or the hormone-positive breast cancer cell line MCF-7 were seeded at $5 \times 10^5$ cells per well in a 6-well tissue culture plate for 24 h.

The cultured supernatant was removed and replaced with fresh RPMI media. To remove dead cells, the cell-free supernatant was collected after 24 h of incubation and centrifuged at $400 \times g$ for 5 min. Fresh CM was prepared for each experiment.

## Quantitative reverse-transcription polymerase chain reaction assay

Total RNA was extracted with the RNeasy Mini kit (QIAGEN, Hilden, Germany). RNA (1μg/sample) was reverse transcribed to yield cDNA using the QuantiTect reverse-transcription kit (QIAGEN). The synthesized cDNA was used as the template to measure the relative expressions of these genes: *CXCL10*, *IL-1B*, *TNF*, *TGFB1*, *TGFBR2*, *CCL22*, *CCL18*, and *IL10*. Quantitative reverse-transcription polymerase chain reaction was performed with StepOnePlusTM (Applied Biosystems, MA, USA) using SYBR green (Applied Biosystems). The mRNA expressions of the candidate genes were normalized to that of the reference gene human *β-actin* and calculated as $2^{-\Delta\Delta Ct}$. The following primers were used in the amplification:

| Genes | | Primer sequence |
|---|---|---|
| *β-actin* | F | 5'-ACCGTGAAAAGATGACCCAG-3' |
| | R | 5'-AGCCTGGATGGCTACGTACA-3' |
| *CXCL10* | F | 5'-GAAAGCAGTTAGCAAGGAAAGGT-3' |
| | R | 5'-GACATATACTCCATGTAGGGAAGTGA-3' |
| *IL1B* | F | 5'-GGCGGCATCCAGCTACGAAT-3' |
| | R | 5'-TCCTGGAAGGTCTGTGGGCA-3' |
| *TNF* | F | 5'-GCATGATCCGGGACGTGGAG-3' |
| | R | 5'-GGGGGCCGATCACTCCAAAG-3' |
| *TGFBR2* | F | 5'-AAGTCGGATGTGGAAATGGA-3' |
| | R | 5'-CAGTGGATGGGCAGTCCTAT-3' |
| *TGFB1* | F | 5'-AGGGCCCAGCATCTGCAAAG-3' |
| | R | 5'-CTGCGTGTCCAGGCTCCAAA-3' |
| *IL10* | F | 5'-TGCCTTCAGCAGAGTGAAGA-3' |
| | R | 5'-GCAACCCAGGTAACCCTTAAA-3' |
| *CCL22* | F | 5'-TCCTTGCTGTGGCGCTTCAA-3' |
| | R | 5'-CTCGGGCAGGAGTCTGAGGT-3' |
| *CCL18* | F | 5'-ATGGCCCTCTGCTCCTGT-3' |
| | R | 5'-AATCTGCCAGGAGGTATAGACG-3' |

## Flow cytometry

THP-1-derived macrophages were washed and resuspended with staining buffer (1X phosphate-buffered saline (PBS) containing 1% FBS). Cell pellets were incubated with anti-CD163 antibody (OTI2G12) (Abcam, Cambridge, UK) for 30 min at 4˚C. Cells were washed and stained with fluorochrome-conjugated secondary antibodies, goat anti-mouse IgG H&L (Alexa Flour® 488, ab150113, Abcam) for 20 min. After staining, all cells were washed twice and analysis was carried out with flow cytometry (BD Accuri™).

## Transwell co-culture assay

THP-1 cells were seeded and stimulated with PMA at a cell density of $5 \times 10^5$ cells/well on Transwell inserts (0.4 μm pore size, Costar 3450; Corning, Inc., NY, USA), followed by a 24-h rest period in fresh RPMI-1640 media. The THP-1 stimulated cells were then incubated with

either IL-4/IL-13 or conditioned media harvested from MDA-MB-231 cells for 48 h to generate cytokine-polarized M2 and TAMs as mentioned earlier. Subsequently, the Transwell inserts containing the THP-1-derived macrophage were placed with MDA-MB-231 cells that were pre-seeded overnight at a cell density of $5 \times 10^5$ cells/well in a Transwell-suitable 6-well plate. After co-culturing for 72 h, the Transwell inserts were removed, and the MDA-MB-231 cells were harvested and subjected to cell proliferation and migration assays.

## Cell proliferation assay

MDA-MB-231 cells were harvested after the Transwell co-culture assay for the measurement of cell proliferation. MDA-MB-231-treated cells were seeded in a 96-well plate at a concentration of $5 \times 10^3$ cells/well in 100 μl of DMEM. The cell proliferation reagent WST-1 (Sigma-Aldrich) was added at 10 μl/well, and the plate was incubated for 4 h. Cell viability was measured at 24, 48, and 72 h, according to the manufacturer's instructions.

## Cell migration assay

MDA-MB-231 cells were harvested after the Transwell co-culture assay and seeded into 96-well plates at a density of $5 \times 10^4$ cells/well. After overnight incubation, a wound was created by scratching the cell monolayer using a 10-μL pipette tip. The remaining cells were washed thoroughly with PBS to remove cellular debris and incubated at 37×C in serum-free media. The wound space was photographed under a phase-contrast microscope (IX51 with DP70, Olympus, Tokyo, Japan) at 0, 6, 12, and 24 h. The total wound area at each time point was determined using ImageJ software and presented as the relative value to 0 h, as previously described [15].

## Cytokine measurement

THP-1-derived macrophages were treated with CM from breast cancer cells or cytokines IL-4/IL-13 for 48 h, as described earlier. The cultured supernatant was removed, washed with PBS, and replaced with fresh RPMI media. After 24 h, to remove dead cells, the cultured supernatant was collected and centrifuged at $400 \times g$ for 5 min. The cultured supernatant collected from THP-1 cells was stimulated with PMA for 24 h, followed by fresh RPMI media for 24 h that served as control. For quantitative detection of cytokine proteins, cell-free supernatant was assayed using the BD™ Cytometric Bead Assay (Becton Dickinson, NJ, USA), including IL-4, IL-10, TNF-α, and IL-6. The assay was performed according to the kit's instruction manual provided by the manufacturer.

## TCGA and METRABRIC Datasets analysis

Gene expression analyses in primary breast cancer tissues and patient clinical data from METABRIC [16] and TCGA [17] were retrieved from cBioPortal [18]. Boxplot analysis was used to compare the mRNA level between two cancer types, TNBC and non-TNBC, categorized by ER, PR, and HER2 status. The number of tissues analyzed in all boxplots was 1,904 in METABRIC (1605 non-TNBC, 299 TNBC) and 1,093 in TCGA (978 non-TNBC, 115 TNBC). Kaplan-Meier survival analysis was performed using METABRIC study. Nine hundred fifty-two patients with available data of gene expression and time to disease-free were included. The mRNA expression was categorized as low and high levels comparing to median expression. Logrank test p-value was used for survival analysis. Cox regression model was used for univariate and multivariate analyses of gene expression level, pathological stage (0–IV), and cancer type (TNBC, non-TNBC). Boxplot and survival analyses were generated using R program.

## Statistical analysis

Results were generated from three or more independent experiments. Values are presented as mean ± standard error of the mean (SEM). Data with three or more groups were analyzed statistically by one-way analysis of variance and Tukey's multiple comparisons test. For comparison between two groups, a two-tailed unpaired t-test was used. A value of $p \leq 0.05$ was accepted as significant (*$p < 0.05$; **, $p < 0.01$; ***, $p < 0.001$; ****, $p < 0.0001$).

# Results

## TNBC cells induce tumor-associated macrophages with characteristics of a mixed M1 and M2 phenotype

TAMs play critical roles in tumor progression. In TNBC, the infiltration of macrophages in the tumor tissue promotes disease progression and leads to a poor prognosis [19]. To examine the crosstalk between TAMs and TNBC cells, the human monocytic leukemic cell line THP-1 was used to generate an in vitro model of TAMs. THP-1 cells were differentiated into macrophages by PMA stimulation for 24 h, followed by treatment with CM obtained from two TNBC cell lines, namely, MDA-MB-231 and MDA-MB-468, to generate TAMs, as described in Fig 1A. As previous observations suggested that most TAMs closely resemble M2 macrophage, PMA-stimulated THP-1 cells were polarized into M2 macrophages with IL-4 and IL-13, as previously described [20]. This population served as a positive control for the comparison of THP-1-differentiated TAMs induced by TNBC-CM. Through flow cytometry, we analyzed the expression of CD163, a surface marker expressed on M2 macrophages. Consistent with previous reports [8], TAMs generated by incubation in TNBC cell CM expressed significantly higher CD163 levels than THP-1-derived macrophages (control). The percentage of CD163+ cells in TNBC-induced TAMs was similar to those of M2-polarized macrophages using IL-4 and IL-13 (Fig 1B and 1C).

We then quantified the mRNA level of classical M1 markers (CXCL10, IL-1β, and TNF-α) and alternative M2 markers (TGF-β1, TGF-βR2, CCL22, CCL18, and IL10) relative to THP-1-derived macrophages (control). Cytokine-polarized M2 macrophages demonstrated a significant upregulation of CCL18 and modest expression of M1-associated genes, confirming the characteristics of the M2 phenotype (Fig 2A and 2B). Interestingly, MDA-MB-231-polarized TAMs exhibited a distinct gene expression profile of M2 phenotype with upregulation of TGF-β1 but no increase in CCL18. Besides, an abundance of M1 associated gene expression including CXCL10, IL-1β, and TNF-α were observed (Fig 2A and 2B). The elongation of macrophages correlates with a reduction of proinflammatory cytokines and polarization of macrophages toward the M2 phenotype [21]. In this study, we observed that M2-polarized THP-1 exhibited pseudopodia formation, whereas unstimulated THP-1 and THP-1-derived macrophages displayed a large, round, single-cell morphology (Fig 2C). However, MDA-MB 231-induced TAMs displayed a mixed morphology of rounded and elongated cells. Moreover, analysis of cytokines secretion revealed that M2-polarized macrophages secreted high levels of IL-4 and low levels of IL-6 and TNF-α, which are M1 mediators that stimulate a proinflammatory immune response [22]. Unlike M2-polarized macrophages, TNBC-induced TAMs secreted a substantial level of IL-10, IL-6, and TNF-α, while IL-4 was barely detected (Fig 2D). Taken together, the results of gene expression, cell morphology, and cytokines secretion have demonstrated that TNBC may induce overlapping and mixed populations of THP-1 derived TAMs. In addition, the mixed M1/M2 phenotype was also observed in primary monocytes that were treated with CM from two different TNBC cell lines; MDA-MB-231 and MDA-MB-468, as indicated by an upregulation of both M2 (CD206) and M1 (CD282, CD284) surface markers (S1 Fig).

A.

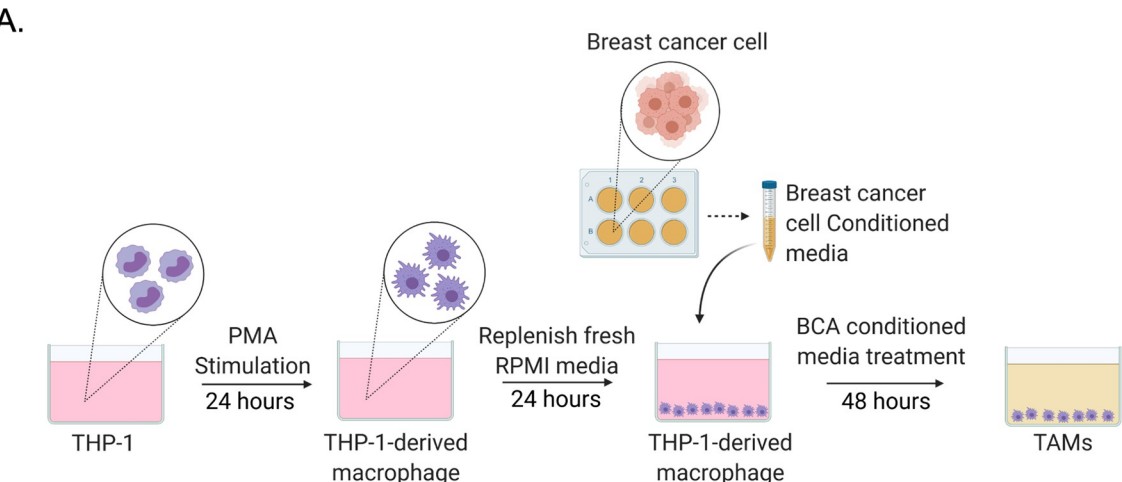

B.

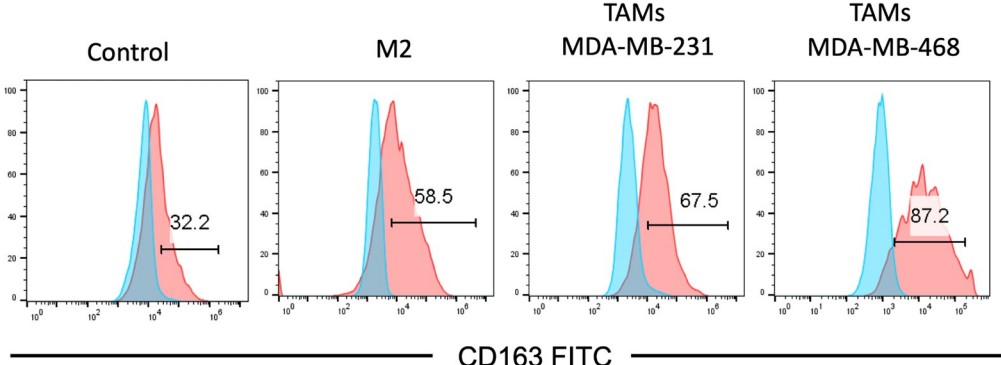

C.

**Fig 1. Generation of TNBC-induced TAMs.** (A) THP-1 cells were PMA-treated for 24 h and washed with RPMI media for another 24 h. The conditioned media collected from breast cancer cells were used to treat differentiated THP-1 and incubated for 48 h to generate TAMs. The illustration was created with BioRender.com. (B) Flow cytometry-based analysis of TAMs and M2-induced THP-1-derived macrophages for the expression of the surface marker CD163 as compared with an isotype control. Control, THP-1-derived macrophages with RPMI media alone; TAMs, THP-1-derived macrophages treated with MDA-MB-231 CM and

MDA-MB-468 CM; and M2, THP-1-derived macrophages treated with IL-4 (20 ng/mL) and IL-13 (20 ng/mL) cytokines. (C) The bar graph represents the mean and SEM from four different experiments with significance level at $^*p < 0.05$, $^*p < 0.05$, $^{**}p < 0.01$, and $^{***}p < 0.001$. TAMs, tumor-associated macrophages; CM, conditioned media; PMA, phorbol 12-myristate 13-acetate; TNBC, triple-negative breast cancer, RPMI, Roswell Park Memorial Institute.

## TNBC-induced TAMs increase breast cancer cell proliferation and migration

In the TNBC microenvironment, tumor stromal cells including macrophages secrete several factors that can promote tumor growth and metastasis [23]. As TNBC-induced TAMs exhibited M1/M2 phenotype heterogeneity and produced the tumor-promoting cytokine IL-6, we sought to evaluate the effects of TNBC-induced TAMs on TNBC growth. THP-1-derived macrophages, M2-polarized macrophages, or MDA-MB-231-induced TAMs were co-cultured with MDA-MB-231 in a Transwell system that allowed the exchange of soluble factors (Fig 3A). After 72 h of co-culture, MDA-MB-231 cells were subjected to WST-1 assay to ascertain their proliferation rate. As displayed in Fig 3B, TAMs with a mixed M1/M2 population significantly enhanced the proliferation of TNBC cells, eliciting a fold change similar to that of M2-polarized macrophages.

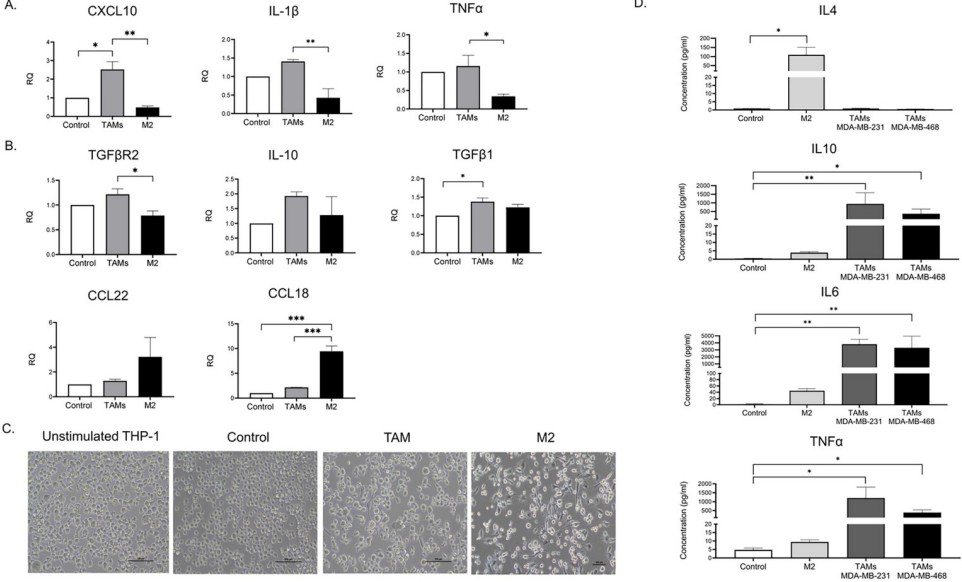

**Fig 2. TNBC-induced TAMs contain a mixed population of M1 and M2.** (A-B) THP-1 derived macrophages were treated with MDA-MB-231 conditioned media (TAMs) or IL-4/IL-13 (M2) for 48 h. The mRNA expression level of (A) M1 markers (CXCL10, IL-1β, and TNF-α) and (B) M2 markers (TGFβ1, TGFβR2, CCL22, CCL18, and IL10) were quantified relative to THP-1-derived macrophages (control). The relative expression level was normalized to the level of the human *β-actin* gene. (C) Morphological observation under microscope of THP-1 cells treated under various conditions; MDA-MB 231 conditioned media (TAMs), IL-4 and IL-13 (M2), PMA stimulated THP-1 alone (control). Bright-field images were displayed at 20× magnification of phase contrast microscopy. Scale bar, 100 μm. (D) The culture supernatant harvested from PMA-stimulated THP-1 treated with MDA-MB 231 conditioned media (TAMs MDA-MB-231), MDA-MB-468 conditioned media (TAMs MDA-MB-468), PMA-stimulated THP-1 treated with IL-4 and IL-13 (M2), and PMA stimulated THP-1 alone (control) were analyzed by CBA assay to measure the concentrations of TNF-α, IL-6, IL-4, and IL-10. Data are represented as the mean ± SEM of three independent experiments, with significance level at $^*p < 0.05$, $^*p < 0.05$, $^{**}p < 0.01$, and $^{***}p < 0.001$. TAMs, tumor-associated macrophages; TNBC, triple-negative breast cancer; IL, interleukin; TNF, tumor necrosis factor; PMA, phorbol 12-myristate 13-acetate.

A.

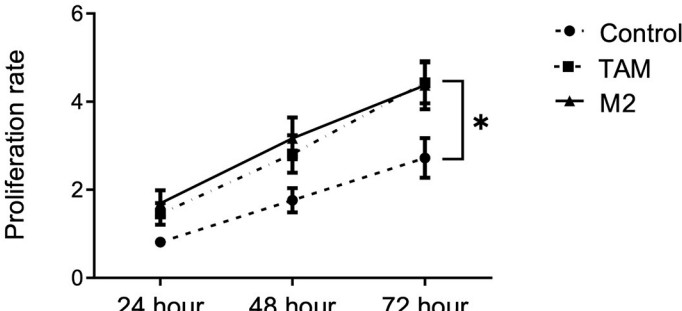

B.

**Fig 3. TAMs increase TNBC cell proliferation.** (A) Schematic diagram of the Transwell assay. THP-1 monocytes were seeded on the Transwell insert and differentiated to macrophage. TAMs were generated by treating THP-1-derived macrophage with CM for 48 h. Transwell inserts containing TAM were co-cultured with MDA-MB-231 seeded in 6-well plate and incubated for 72 h. (B) Cell proliferation rate evaluated by WST-1 assay. Data are calculated as fold change to time 0 and represented as mean ± SEM of 3–4 independent experiments, with significance level at *p < 0.05. TAMs, tumor-associated macrophages; TNBC, triple-negative breast cancer; CM, conditioned media.

Then, we determined whether TAMs could promote TNBC cell migration, which is a critical step in tumor metastasis. We used a wound-healing assay to determine the total wound area at each time point, as previously described [15]. The migratory activity of MDA-MB-231 was measured at 6, 12, and 24 h after the infliction of a scratch wound to exclude the cell proliferation effect (Fig 4A). TAMs generated from MDA-MB-231-CM and cytokine-polarized M2 macrophages significantly increased the mobility of TNBC cells (Fig 4A and 4B). Thus, our results suggest that TAMs with a mixed M1/M2 phenotype could promote cell proliferation and migration in TNBC cells similar to that of IL-4/IL-13-polarized M2 macrophages.

## TNBC but not HR⁺ BCA secrete high amounts of IL6

A study revealed that BCA with different molecular characteristics distinctly influenced macrophage phenotype and function [10]. Thus, we sought to investigate the phenotype of TAMs induced by hormone receptor-positive (HR⁺) BCA. We observed that TAMs polarized by CM collected from HR⁺ BCA (MCF-7 cells) demonstrated an upregulation of M2-associated genes, including *TGF-b1* and *CCL22*. However, M1-associated proinflammatory mediator

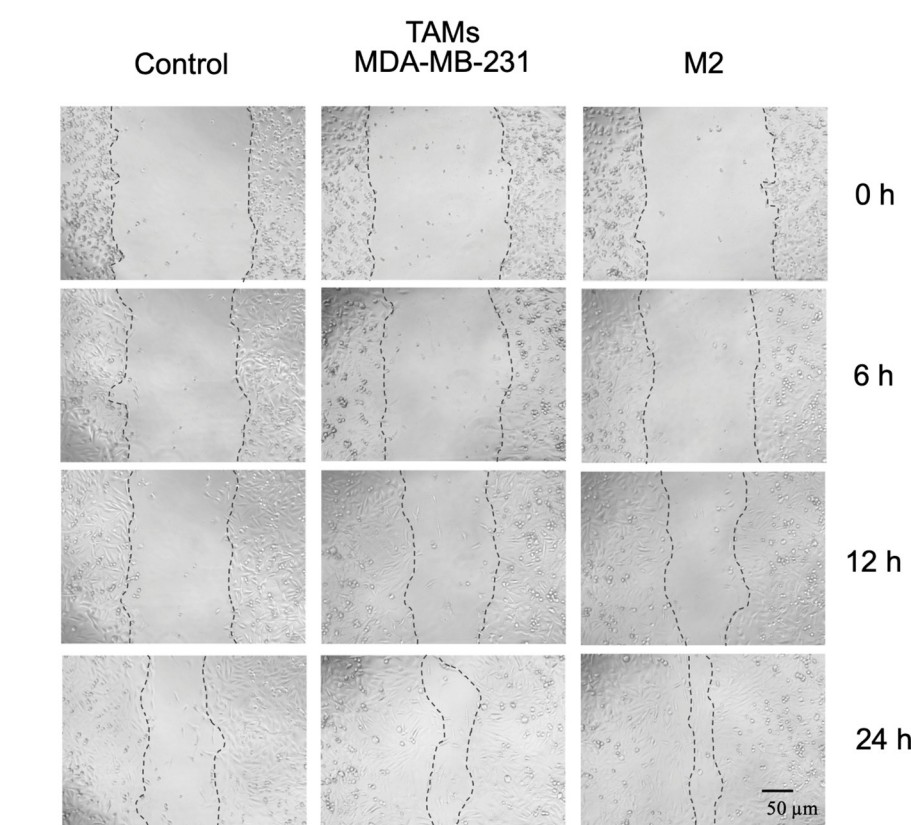

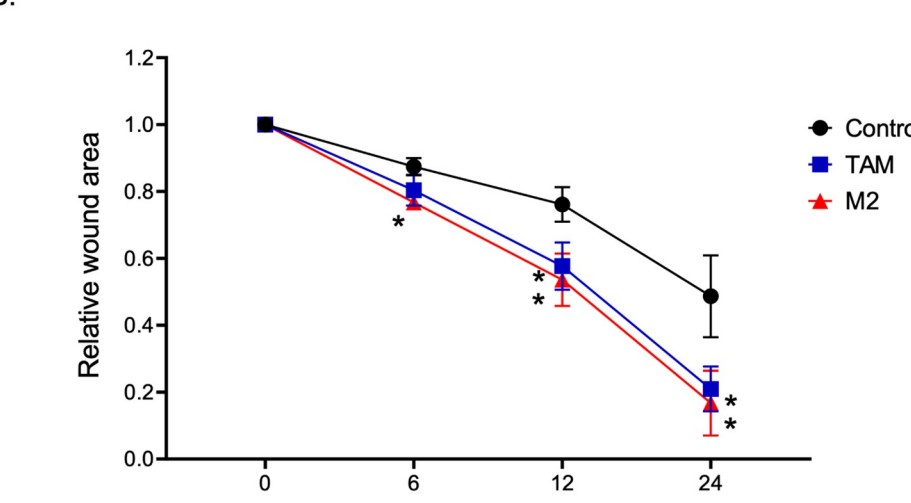

**Fig 4. TAM promotes the migratory activity of TNBC cells.** (A) Representative images of the wound migration assay of MDA-MB-231 cells measured at 0 h, 6 h, 12 h, and 24 h under a phase-contrast microscope. (B) Time course of wound closures expressed as the remaining wound area relative to time point 0 h. Data are represented as mean ± SEM of 3–4 independent experiments, with significance level at *p < 0.05 vs control. TAMs, tumor-associated macrophages; TNBC, triple-negative breast cancer.

genes including *CXCL10*, *IL-1b*, and *TNF-a* were significantly downregulated in MCF-7-induced TAMs (Fig 5A). Compared with TNBC-induced TAMs and IL-4/IL-13-polarized M2 macrophages, the cytokine gene profile of HR⁺-induced TAMs was related to those of cytokine-polarized macrophages, whereas TNBC-TAMs exhibited a unique differentially

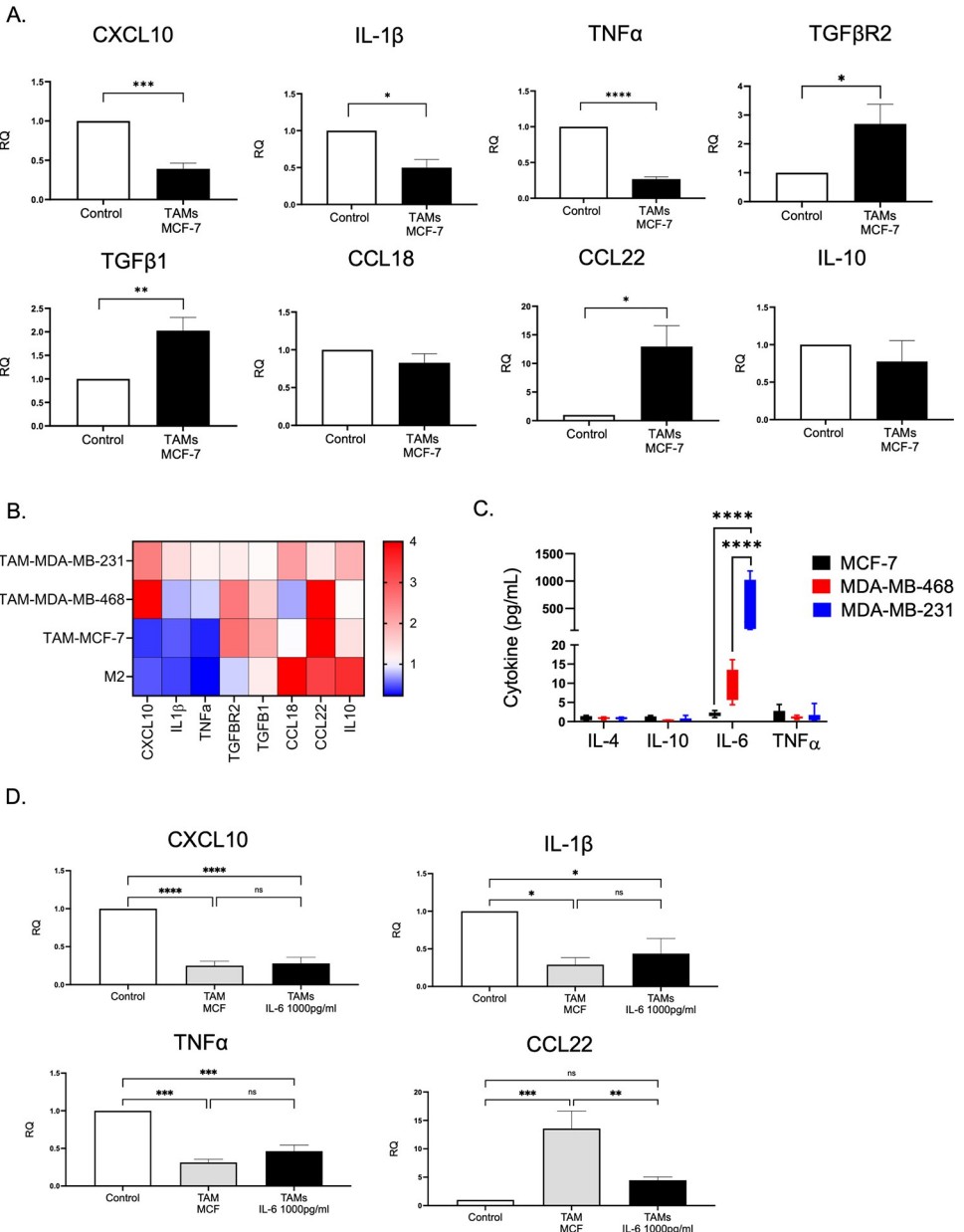

**Fig 5. TNBC but not HR⁺ BCA secrete high amounts of IL6.** (A) mRNA expression levels of M1 markers (CXCL10, IL-1β, and TNF-α) and M2 markers (TGFβ1, TGFβR2, CCL18, CCL22, and IL10) of MCF-7 CM-induced TAMs. (B) Heat map of the mRNA expression levels of TAMs induced by the conditioned media from MDA-MB-231, MDA-MB-468, or MCF-7, and IL-4/IL-13 polarized macrophages (M2). (C) Quantitative detection of cytokines IL4, IL10, IL6, and TNF-α from the conditioned-media of MDA-MB-231, MDA-MB-468, and MCF-7 using CBA assay. (D) The mRNA expression levels of CXCL10, IL-1β, TNF-α, and CCL22 were measured in TAMs generated from MCF-7-CM supplemented with rhIL-6 1000 pg/mL. Data are represented as mean ± SEM of three independent experiments, with significance level at *p < 0.05, **p < 0.01, ***p < 0.001, ****, and p < 0.0001. TAMs, tumor-associated macrophages; TNBC, triple-negative breast cancer; CM, conditioned media; TNF, tumor necrosis factor; IL, interleukin.

expressed gene profile (Fig 5B). These results support previous evidence that TNBC regulated distinct macrophage phenotype and biological response.

Then, we determined the cytokine secretion profile in TNBC and HR⁺ BCA cell lines. Cultured supernatants from MDA-MB-231, MDA-MB-468, and MCF-7 cells were collected and the levels of IL-4, IL-10, IL-6, and TNF-α were quantified. MDA-MB-231 cells secreted large amounts of IL-6 compared with MDA-MB-468 and MCF-7 cells. The level of secreted IL-6 was also significantly higher in MDA-MB-468 cells than in HR⁺ MCF-7 cells. We found that the levels of IL-4, IL-10, and TNF-α were not significantly different among the three cell lines (Fig 5C). To address the possibility that a high IL-6 level may contribute to the M1/M2 mixed phenotype of TAMs educated by MDA-MB-231, we polarized PMA-stimulated THP-1 with MCF-7-CM in the presence of rhIL-6 at 1000 pg/mL and quantified the mRNA level of related genes. We hypothesized that the addition of exogenous rhIL-6 may result in the upregulation of proinflammatory cytokines in MCF-7 induced TAMs. However, only a significant reduction of CCL22 gene expression with the addition of IL-6 in the MCF-CM polarized condition was observed, but the mRNA expressions of M1-associated genes including *CXCL10*, *IL1B*, and *TNF* were not different between MCF-7-TAMs and MCF-7-TAMs supplemented with rhIL-6.

## M1-associated genes are upregulated in human TNBC tissues

Collectively, our data demonstrate the upregulation of M1-associated genes in TNBC induced TAMs but not in the HR⁺ BCA cells. To consider whether this phenomenon correlate to breast cancer environment in humans, differential gene expression analyses of M1- and M2-associated genes were analyzed in TNBC versus non-TNBC tissues from patients using two independent public databases, METRABRIC and TCGA (Fig 6A). Three M1-associated genes, including *CXCL10*, *IL1B*, *and TNF* were significantly upregulated in TNBC tissues in both datasets. Additional cytokine genes including *IL-4*, *IL-10*, and *IL-6* were also analyzed. In consistency with the in vitro result, *IL6* showed significantly high level in TNBC comparing to non-TNBC tissues. Furthermore, we performed Kaplan-Meier survival analyses of patients bearing high versus low levels of M1-associated genes (Fig 6B). Interestingly, high *CXCL10* level was significantly associated with shorter disease-free period, whereas high *IL1B* level was associated with delayed disease progression. Univariate and multivariate analyses were performed with tumor stage and cancer type as covariates (Table 1). The results showed that expression levels of *CXCL10* and *IL1B* were significantly prognostic of disease-free survival independently of other clinical factors.

## Discussion

In the tumor microenvironment, TAMs, originating from blood monocytes, are recruited to tumor tissues by factors secreted by tumor cells and the tumor stroma and constitute the majority of tumor-infiltrating immune cells. Given the high plasticity and diversity of myeloid-lineage cells, the tumor milieu, depending on the location, tumor type, and local stromal cells, shapes the polarization and biological function of TAMs [24–26]. By contrast, TAM infiltration is a poor prognostic factor and associated with aggressive tumor characteristics [27–29]. This crosstalk between cancer cells and TAMs is widely validated in ex vivo experiments where human monocytic leukemic cells (THP-1) were co-cultured with cancer cells or cancer CM.

THP-1 monocytes can be differentiated in vitro using PMA into macrophage-like cells that resemble functional mature macrophages [30–32]. Genin et al. reported that PMA-differentiated THP-1 can be polarized into M1 or M2 macrophages with expression profiles similar to

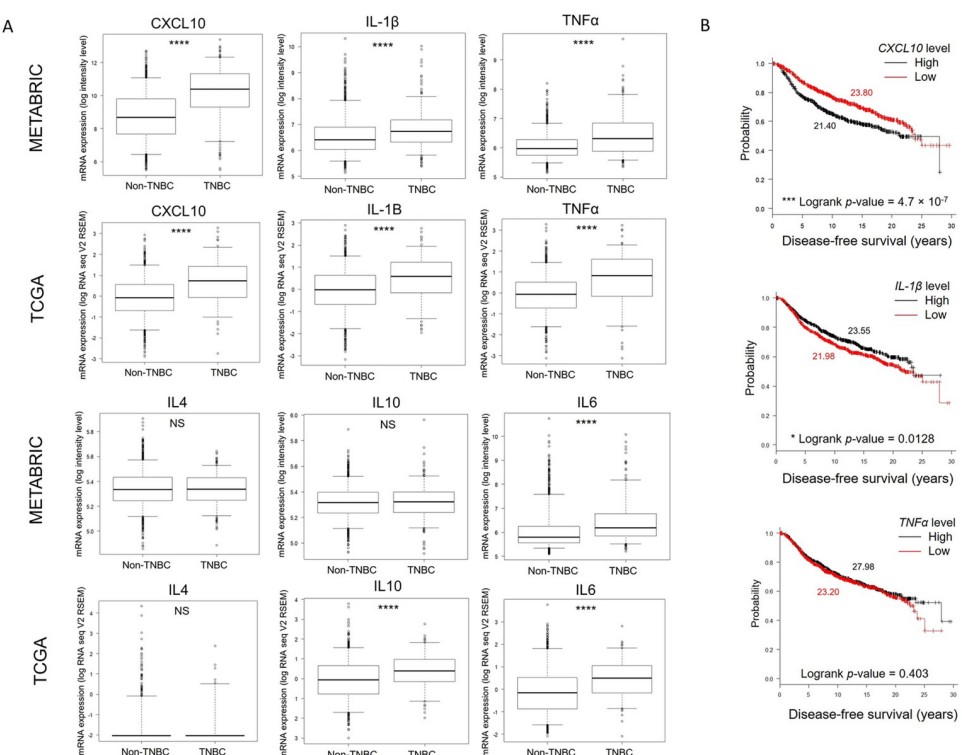

**Fig 6. Upregulation of M1-associated genes in human breast cancer.** (A) The differential gene expression analyses of M1- and M2-associated genes in tumor tissues from patients with breast carcinoma retrieved from METABRIC and TCGA studies. Boxplot analyses of indicated genes in TNBC versus non-TNBC. Mann–Whitney U-test, two-sided p-value; ****p < 0.0001; *p < 0.05; NS, p ≥ 0.05. (B) Kaplan-Meier survival curves of indicated genes categorized as low and high mRNA levels comparing to median expression. Median disease-free survival (red, low; black, high) and Logrank p-value are showed.

those of polarized primary monocytes. Moreover, these THP-1 M2-like macrophages prevented an apoptotic response to etoposide in two cancer cell lines, namely, HepG2 and A549 [20]. Multiple evidence also indicates that CM from cancer cells regulate the differentiation of THP-1 monocytes into tumor-promoting M2-like macrophages [33,34].

In breast cancer, THP-1 treated with CM from MCF-7 or MDA-MB 231 cells results in the differentiation of THP-1 cells toward macrophages and increases the mRNA expression of M2-type macrophage markers, including CD163 and IL-10 [35]. In this study, we consistently reported that CM from MDA-MB-231 cells induced PMA-stimulated THP-1 to express

**Table 1. Univariate and multivariate cox regression model using METABRIC database.**

| Variables | Reference | Univariate analysis | | Multivariate analysis | |
|---|---|---|---|---|---|
| | | HR (95% CI) | *P*-value | HR (95% CI) | *P*-value |
| *CXCL10* Level (≥ median) | Low (< median) | 0.67 (0.57–0.78) | $5.63 \times 10^{-7}$ | 0.70 (0.57–0.85) | $4.42 \times 10^{-4}$ |
| *IL1B* Level (≥ median) | Low (< median) | 1.22 (1.04–1.43) | 0.0130 | 1.31 (1.09–1.60) | 0.0045 |
| *TNF* Level (≥ median) | Low (< median) | 1.07 (0.91–1.25) | 0.4034 | 1.15 (0.95–1.39) | 0.1551 |
| Tumor stage (I–VI) | Stage 0 | 2.12 (1.84–2.44) | < 0.0000 | 2.02 (1.75–2.32) | < 0.0000 |
| Cancer type (TNBC) | Non-TNBC | 1.37 (1.12–1.68) | 0.0025 | 1.18 (0.92–1.51) | 0.1885 |

HR, hazard ratio; CI, confidence interval; Each variable was analyzed in 1,904 breast cancer patients.

CD163 on the cell surface and upregulate the immunosuppressive cytokine TGFβ1 and its cognate receptor TGFβR2. In contrast to THP-1-derived macrophages polarized by IL-4/IL-13, TNBC-induced TAMs demonstrated high expressions of M1-associated genes, including that of *CXCL10*, *IL-1b* and *TNF-a*. We also observed less pseudopodia formation in TNBC-induced TAMs than in IL-4/IL-13-polarized M2 macrophages. Our results indicate that soluble factors secreted by TNBC cells induce the polarization of THP-1-derived macrophages into a distinct TAM population with defining characteristics of M1 and M2 types. This phenomenon appears to be preserved in TNBCs because we did not observe the upregulation of M1-associated genes in HR[+] MCF-7 induced TAMs. This is in agreement with the finding of a previous study that THP-1 cells exposed to basal-like breast cancer cells in a co-culture system exhibit the upregulated gene expression of both M1 and M2 macrophage markers [36]. More importantly, Hollmen et al. [2015] performed a whole-transcriptome sequencing of human monocytes co-cultured with MDA-MB-231 or T47D [ER[+]] BCA cells and revealed that different BCA types distinctly educate macrophage phenotypes and functions. ER[+] BCA-activated macrophages display upregulation of acute-phase inflammatory signal, IL-17 signaling, while TNBC-activated macrophages exhibit downregulation of the citrulline pathway associated with nitric oxide [NO] production [10]. The mixed polarization of M1/M2 type in THP-1 treated with tumor-soluble factors was also observed in colorectal cancer, in which CM from several colon cancer cell lines enhanced the phagocytic activity of THP-1 cells [25] and induced the production of cytokines and chemokines typical of both M1 and M2 macrophages, including IL-1β, MCP-1, IL-6, and IL-10 [37].

At the protein level, we reported that TNBC-induced TAMs but not M2-polarized macrophage secrete large amounts of IL-6 and TNF-α. The abundance of IL-6 and TNF-α was comparable between MDA-MB-231 and MDA-MB-468 cells. TNBC-induced TAMs also secrete high levels of IL-10 but not IL-4. A study revealed that TAMs infiltrating TNBCs secrete IL-10, which contributes to tumor progression [38]. In our study, M2-polarized macrophages secreted significant amounts of IL-4, which are proposed to possess anti-inflammatory and tumor-promoting properties [39]. However, only a small amount of IL-10 in the culture supernatant of M2-polarized macrophages was detected. We examined the CM of MDA-MB-231 and detected a very small amount of IL-4. This absence of IL-4 in MDA-MB-231-CM could be attributed to the different phenotypes of TNBC-activated TAMs from M2-macrophages polarized by IL-4/IL-13. Chan et al. analyzed cytokine expression using MILLIPLEX assay in human breast cancer cell lines and reported that MDA- MB-231 cells secreted moderate and low amounts of IL-4 and IL-13, respectively [40].

In line with previous findings [40], TNBC cell lines significantly secreted higher amounts of IL-6 than did the HR[+] BCA cell line. The magnitude of IL-6 production by TNBC cells is correlated with aggressive behavior, where MDA-MB-468, a less aggressive cell line, secretes a much lower level of IL-6 than MDA-MB-231. Several tumor cells, including breast cancer cells, can secrete IL-6, the upregulation of which is generally associated with poor prognosis and a low survival outcome [41,42]. Previous evidence also revealed IL-6 promotes TNBC cell survival [43], increases breast cancer stemness [44], and contributes to chemoresistance of MDA-MB 231 cells [45]. Furthermore, primary macrophages secrete IL-6, which in turn increase IL-6 secretion from cancer cells [46], where IL-6 is shown to perpetuate this vicious cycle by generating more aggressive M2 macrophage polarization by activating Stat3 phosphorylation [44,47]. In the present study, we investigated the influence of IL-6 on the distinct characteristics of TNBC-induced TAMs by the generation of TAMs with MCF-7-CM-containing rhIL-6 at a similar concentration found in the MDA-MB-231-CM. However, the addition of IL-6 did not upregulate M1-associated genes in MCF-7-induced TAMs, suggesting that multiple mediators contribute to the complexity of macrophage polarization induced by breast

cancer. Interestingly, the incorporation of IL-6 downregulates *CCL22* in MCF-7-induced TAMs. *CCL22* is expressed in alternative (M2) macrophage [48,49] and in many types of cancer [50–52]. The functional role of CCL22 in cancer immunity involves recruiting Tregs to the tumor stroma by binding to its cognate receptor CCR4 [53,54]. A previous study suggested that CCL22 is downregulated by the TH1 cytokine IFNγ [55]. The mechanistic interaction between IL-6 and CCL22 in TNBC should be explored for further investigation. Future experiments such as the incorporation of IL-6 neutralizing antibody or the deletion of IL-6 with siRNA could be necessary to determine the role of IL-6 in the polarization of TNBC-educated TAMs.

Finally, the results from in vitro experiments were validated by analyses of two independent patient databases that showed a significant greater M1-associated genes expression in TNBC than non-TNBC. The expression level of *IL6* was also significantly higher in TNBC compared with non-TNBC tissues, whereas *IL4* level was not different between cancer types. This result supported our finding that TNBC cells secreted large amounts of IL-6 comparing to non-TNBC cells. Interestingly, the expression of *CXCL10* have strong prognostic value for disease-free survival in the breast cancer human cohort, suggesting a critical contribution of this molecule in breast cancer outcomes.

In summary, our results demonstrate that TNBC cells but not HR+ cells induce a distinct population of M2-like TAMs that express M1-associated genes and secrete a considerable amount of IL-10 and IL-6, which in turn promotes breast cancer cell proliferation and metastasis. Although these TNBC-induced TAMs exhibit a differential gene expression profile, their tumor-promoting effects were similar to those of M2 macrophages polarized by IL-4 and IL-13. One of the limitations in this study is the lack of M1 and M2 associated enzymes and proteins analyses. Nevertheless, the M1/M2 mixed phenotype on TNBC-induced TAMs were demonstrated by gene expression, cellular morphology, and cytokines profile of THP-1 derived macrophage as well as surface expression of M1 and M2 markers on primary monocytes. Moreover, analysis of patient database follows the in vitro finding where M1 associated genes are upregulated in TNBCs particularly the key marker CXCL-10 which is negatively correlated with disease-free survival. Together, our results suggest the complexity and heterogeneity of TAMs and their negative roles in TNBC.

## Supporting information

**S1 Fig. TNBC conditioned media induces mixed phenotype of primary human monocytes.** Monocyte are isolated from PBMC using CD14 magnetic bead. Primary CD14+ monocytes were incubated with RPMI media only (control), MDA-MB 231 conditioned media (TAMs MDA-MB-231), MDA-MB-468 conditioned media (TAMs MDA-MB-468), or IL-4 (20 ng/mL) and IL-13 (20 ng/mL) cytokines (M2) for 48 h. (A) Monocyte-derived macrophages expression level of M1 (CD282 and CD284) and M2 (CD206) surface markers were analyzed by flow cytometry as compared with an isotype control. (B) The bar graph represents the mean and SEM of M1 and M2 markers from three different experiments with significance level at $^*$p < 0.05, $^*$p < 0.05, $^{**}$p < 0.01, and $^{***}$p < 0.001.
(TIF)

**S1 File. Methodology.**
(DOCX)

## Author Contributions

**Conceptualization:** Koramit Suppipat, Pithi Chanvorachote, Supannikar Tawinwung.

**Data curation:** Kristine Cate S. Pe, Supannikar Tawinwung.

**Formal analysis:** Kristine Cate S. Pe, Rattana Saetung, Varalee Yodsurang, Chatchai Chaotham, Supannikar Tawinwung.

**Funding acquisition:** Pithi Chanvorachote, Supannikar Tawinwung.

**Investigation:** Kristine Cate S. Pe, Rattana Saetung, Varalee Yodsurang, Chatchai Chaotham.

**Methodology:** Koramit Suppipat, Supannikar Tawinwung.

**Project administration:** Supannikar Tawinwung.

**Resources:** Koramit Suppipat, Supannikar Tawinwung.

**Supervision:** Koramit Suppipat, Pithi Chanvorachote, Supannikar Tawinwung.

**Visualization:** Kristine Cate S. Pe, Varalee Yodsurang.

**Writing – original draft:** Kristine Cate S. Pe, Varalee Yodsurang, Chatchai Chaotham, Supannikar Tawinwung.

**Writing – review & editing:** Varalee Yodsurang, Chatchai Chaotham, Koramit Suppipat, Supannikar Tawinwung.

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
