## [Decision Letter · Decision Letter 0]

4 Apr 2022

PONE-D-21-39755Triple-negative breast cancer influences a mixed M1/M2 macrophage phenotype associated with tumor aggressivenessPLOS ONE

Dear Dr. Supannikar Tawinwung, 

Thank you for submitting your manuscript to PLOS ONE. After careful consideration, we feel that it has merit but does not fully meet PLOS ONE’s publication criteria as it currently stands. Therefore, we invite you to submit a revised version of the manuscript that addresses the points raised during the review process.

Data obtained with THP-1 cells should be validated with PBMC-derived primary macrophages which will be helpful to the improvement of the manuscript, in terms of clarifying the methodology and the goal of this manuscript. The authors should also show if such M1-associated genes induced by conditioned THP-1 cells is accompanied by expression of classical M1 surface markers. The authors also need to address  the protein level after stimulation of THP-1 cells with CM medium through western blot. The manuscript should be checked for typos and grammar errors.

We would appreciate receiving your revised manuscript by  May 19 2022 11:59PM. If you will need more time than this to complete your revisions, please reply to this message or contact the journal office at plosone@plos.org. Please include the following items when submitting your revised manuscript:A rebuttal letter that responds to each point raised by the academic editor and reviewer(s). You should upload this letter as a separate file labeled 'Response to Reviewers'.A marked-up copy of your manuscript that highlights changes made to the original version. You should upload this as a separate file labeled 'Revised Manuscript with Track Changes'.An unmarked version of your revised paper without tracked changes. You should upload this as a separate file labeled 'Manuscript'.

We look forward to receiving your revised manuscript.

Kind regards,

Habib Boukerche, PhD

Academic Editor

PLOS ONE

Journal Requirements:

**
*Reviewers' comments:*
**

**
Reviewer's Responses to Questions
**

**Comments to the Author**

1. Is the manuscript technically sound, and do the data support the conclusions?

Reviewer #1: Partly

Reviewer #2: Yes

2. Has the statistical analysis been performed appropriately and rigorously? 

Reviewer #1: Yes

Reviewer #2: Yes

3. Have the authors made all data underlying the findings in their manuscript fully available?

Reviewer #1: Yes

Reviewer #2: Yes

4. Is the manuscript presented in an intelligible fashion and written in standard English?

Reviewer #1: Yes

Reviewer #2: Yes

5. Review Comments to the Author

**Reviewer #1:** General comments

In this manuscript the authors attempt to characterize an in-vitro THP-1 cell -based macrophage model exposed to conditioned media from TNBC, trying to validate the results in specific patients’ datasets. Their conclusion is that the obtained TNBC conditioned THP-1 cells phenotype is mixed between the M1- and M2 -like one, suggesting the heterogeneity of TAM phenotype in breast cancer.

In my opinion the manuscript cannot be accepted and requires major revisions.

As a general comment, I find it difficult to accept that a tumor cell line (THP-1 cells) conditioned with TNBC media could be compared to TAMs. At least the results obtained with THP-1 cells should be validated with PBMC-derived primary macrophages.

First of all, conditioned THP-1 cells express M1-associated genes without reporting the indicated markers from a classically-stimulated positive control

Except for CXCL10 and TGF1B, ThP-1 conditioning does not induce a significant increase for any of the markers indicated in Figure 2, when compared to the corresponding control. So it is not clear if these conditioned cells have reached their polarization status or not. Then Figure 3 shows a relevant increase in the IL-6 , IL-10 and TNFa release by TNBC-conditioned cells, so these data should be included in Figure 2. It is also not clear in Figure 2 the media of which cancer cell type was used to condition ThP-1 cells.

Additionally, ThP-1 alternative polarization (labelled as M2, obtained with stimulation with IL-4+IL13 I believe) does not induce a significant increase of the typical associated markers (Figure 2), which is also quite strange. Since IL-4 and IL-13 produce different effects, the indicated markers should be evaluated in cells stimulated with IL-4 alone and IL-13 alone. Additionally, data regarding the expression of M1 and M2-related metabolic enzymes, such as iNOS, PKM2 or ARG1 should also be included.

To evaluate the functional features of these conditioned ThP-1 cells, some migration experiments were performed finding that TAMs and M2-polarized macrophages increased the motility of TBNC cells. In the same way, the authors should study the capacity of these conditioned cells to increase the invasive properties of TBNC cells and analyze the expression levels of classical migration/invasion markers such as N-cadherin, E-cadherin or vimentin. Also, since these cells exhibit a significant increase in the CXCL10 expression levels, the ability of these cells to induce CD8+ T cell recruitment should be reported.

Specific comments

Page 3, line 40 -42 � In this paragraph this sentence is written: “Accumulated evidence suggests that TAMs are a heterogeneous and plastic population, in which polarized TAMs can be identified as M1- and M2-like macrophages”. Please reference.

6. Line 219 � Please write the full name for the acronym BCA.

Figure legends: please specify the cell types and the treatment each result comes from.

Methods: please describe the transwell assay with more detail.

**Reviewer #2**: This manuscript is well written and performed experiments well, after a minor revision manuscript could be accepted Manuscript Title: "Triple-negative breast cancer influences a mixed M1/M2 macrophage phenotype associated with tumor aggressiveness."

Manuscript is written well and informative way. Data were clean and clear and followed the flow of writing.

I wonder that if authors also need to consider the protein level after stimulation of THP-1 cells with CM medium through western blotting which will tell more profound story of the manuscript.

I recommend that if authors could perform major proteins western blotting and showed the phenotypic condition in protein level that would be more readable.

After the minor revision of experiments this manuscript could be accepted in this journal

6. PLOS authors have the option to publish the peer review history of their article (what does this mean?). If published, this will include your full peer review and any attached files.

Reviewer #1: No

Reviewer #2: **Yes: **Yuba Raj Pokharel

---

## [Author Response · Author response to Decision Letter 0]

16 May 2022

Editor’s comments

Comment #1: Data obtained with THP-1 cells should be validated with PBMC-derived primary macrophages which will be helpful to the improvement of the manuscript, in terms of clarifying the methodology and the goal of this manuscript. 

Answer: We have performed an experiment on primary monocytes as editor’s suggestion. In the addition experiment, primary monocytes were treated with either conditioned media from two different TNBC cell lines or cytokines IL-4 and IL-13 for M2 induction, similarly to the THP-1 condition. We used flow cytometry to detect surface expression of M1 and M2 markers including TLR-2 (CD282) and TLR-4 (CD284) as M1 markers and CD206 for M2 marker. The results are described in the revised manuscript at page 10, line 235-240 and in a supplementary figure 1.

Comment #2: The authors should also show if such M1-associated genes induced by conditioned THP-1 cells is accompanied by expression of classical M1 surface markers. The authors also need to address the protein level after stimulation of THP-1 cells with CM medium through western blot. 

Answer: We agree that the incorporation of protein analysis would make the finding more profound. However, due to our laboratory limitation, we could not perform such experiments as recommended. Therefore, we have mentioned this limitation in the revised manuscript on page 19, line 467-474. Also, the M1/M2 mixed phenotype on TNBC-induced TAMs in our study were further verified by surface expression of M1 and M2 markers on primary monocytes that were conditioned with TNBC cultured media. Moreover, analysis of two independent patient database follows our in vitro finding where M1 associated genes are upregulated in TNBCs particularly the key marker CXCL-10 which is negatively correlated with disease-free survival.

Reviewer 1’s comments

Comment#1: As a general comment, I find it difficult to accept that a tumor cell line (THP-1 cells) conditioned with TNBC media could be compared to TAMs. At least the results obtained with THP-1 cells should be validated with PBMC-derived primary macrophages.

Answer: As mentioned earlier, we have performed an experiment on primary monocytes as recommended. The results are described in the revised manuscript at page 10, line 235-240 and in a supplementary figure 1.

Comment#2: First of all, conditioned THP-1 cells express M1-associated genes without reporting the indicated markers from a classically-stimulated positive control. Except for CXCL10 and TGF1B, ThP-1 conditioning does not induce a significant increase for any of the markers indicated in Figure 2, when compared to the corresponding control. So it is not clear if these conditioned cells have reached their polarization status or not. Then Figure 3 shows a relevant increase in the IL-6 , IL-10 and TNFa release by TNBC-conditioned cells, so these data should be included in Figure 2. 

Answer: In addition to M1 and M2 associated gene expression, the characterization of TNBC-conditioned cells was also determined with cellular morphology and secreted cytokines. In the revised manuscript, we have combined the figure 2 and 3 as suggested. 

Comment3#: It is also not clear in Figure 2 the media of which cancer cell type was used to condition ThP-1 cells.

Answer: The conditioned media from MDA-MB-231 cells were used to treat PMA-stimulated THP-1 for M1- and M2- associated gene expression and morphological observation. This was also clarified in the figure legend of figure 2 in the revised manuscript.

Comment 4# Additionally, ThP-1 alternative polarization (labelled as M2, obtained with stimulation with IL-4+IL13 I believe) does not induce a significant increase of the typical associated markers (Figure 2), which is also quite strange. Since IL-4 and IL-13 produce different effects, the indicated markers should be evaluated in cells stimulated with IL-4 alone and IL-13 alone. 

Answer: We used the combination of IL-4 and IL-13 to polarize M2 macrophage based on the earlier report by Genin et al. (BMC Cancer volume 15, Article number: 577 (2015)) in which we consistently observed the upregulation of CCL22 and CCL18. The extent of increase, however, is different between our study and theirs. The different stimulation time, passages of cell lines and culture conditions could attribute to this discrepancy between studies. 

Comment #5: Additionally, data regarding the expression of M1 and M2-related metabolic enzymes, such as iNOS, PKM2 or ARG1 should also be included.

Answer: Although, in-dept characterization of classical M1 and M2 macrophage would be ideal for the current manuscript, the primary objective of the current study is to demonstrate the complexity and heterogeneity of TAMs induced by TNBC which were evidenced by common M1 and M2 gene expression, morphology, and secreted mediators as well as the confirmation of surface markers in primary monocytes. 

Comment #6: To evaluate the functional features of these conditioned ThP-1 cells, some migration experiments were performed finding that TAMs and M2-polarized macrophages increased the motility of TBNC cells. In the same way, the authors should study the capacity of these conditioned cells to increase the invasive properties of TBNC cells and analyze the expression levels of classical migration/invasion markers such as N-cadherin, E-cadherin or vimentin. 

Answer: We have used migration and proliferation assay as a functional study to confirm the negative effects of our TNBC-induced TAMs model on breast cancer. Previous studies have also shown the effects of TAMs on the increased expression of migration/invasion markers on various types of cancer (Gan et al., 2016, Wei et al., 2019, Yao et al., 2018)

Comment #7: Also, since these cells exhibit a significant increase in the CXCL10 expression levels, the ability of these cells to induce CD8+ T cell recruitment should be reported.

Answer: The effect of TAMs to immune cells is out of scope for this manuscript.

Comment #8: Page 3, line 40 -42, In this paragraph this sentence is written: “Accumulated evidence suggests that TAMs are a heterogeneous and plastic population, in which polarized TAMs can be identified as M1- and M2-like macrophages”. Please reference.

Answer: Reference were done. Page 3, line 48

Comment #9: Line 219 Please write the full name for the acronym BCA.

Answer: Correction has been done. Page 11, line 264

Comment #10: Figure legends: please specify the cell types and the treatment each result comes from.

Answer: The figure legend has been revised. 

Comment #11: Methods: please describe the transwell assay with more detail.

Answer: More description of the Transwell assay has been provided. Page 6 Line 118-127

Reviewer 2’s comments

Comment#1: Manuscript is written well and informative way. Data were clean and clear and followed the flow of writing. I wonder that if authors also need to consider the protein level after stimulation of THP-1 cells with CM medium through western blotting which will tell more profound story of the manuscript. I recommend that if authors could perform major proteins western blotting and showed the phenotypic condition in protein level that would be more readable.

Answer: We agree that the incorporation of protein analysis would make the finding more profound. However, due to our laboratory limitation, we could not perform such experiments as recommended. Therefore, we have mentioned this limitation in the revised manuscript on page 19, line 467-474. The M1/M2 mixed phenotype on TNBC-induced TAMs in our study were verified by gene expression, morphology, and cytokines profile of THP-1 derived macrophage as well as surface expression of M1 and M2 markers on primary monocytes. Moreover, analysis of two independent patient database follows our in vitro finding where M1 associated genes are upregulated in TNBCs particularly the key marker CXCL-10 which is negatively correlated with disease-free survival.

---

## [Editor Report · Decision Letter 1]

2 Aug 2022

Triple-negative breast cancer influences a mixed M1/M2 macrophage phenotype associated with tumor aggressiveness

PONE-D-21-39755R1

Dear Dr. Tawinwung,

We’re pleased to inform you that your manuscript has been judged scientifically suitable for publication and will be formally accepted for publication once it meets all outstanding technical requirements.

Kind regards,

Dominique Heymann, Ph.D.

Academic Editor

PLOS ONE
---

## [Editor Report · Acceptance letter]

4 Aug 2022

PONE-D-21-39755R1 

*Triple-negative breast cancer influences a mixed M1/M2 macrophage phenotype associated with tumor aggressiveness*

Dear Dr. Tawinwung:

I'm pleased to inform you that your manuscript has been deemed suitable for publication in PLOS ONE. Congratulations! Your manuscript is now with our production department. 

Kind regards, 

on behalf of

Pr. Dominique Heymann 

Academic Editor

PLOS ONE